# Transcriptional dissection of symptomatic profiles across the brain of men and women with depression

Samaneh Mansouri[1,2], André M. Pessoni [1,3], Arturo Marroquín-Rivera[1,3], Eric M. Parise [4], Carol A. Tamminga [5], Gustavo Turecki [6,7], Eric J. Nestler [4], Ting-Huei Chen[1,8] & Benoit Labonté [1,3] ✉

Major depressive disorder (MDD) is one of the most important causes of disability worldwide. While recent work provides insights into the molecular alterations in the brain of patients with MDD, whether these molecular signatures can be associated with the expression of specific symptom domains remains unclear. Here, we identified sex-specific gene modules associated with the expression of MDD, combining differential gene expression and co-expression network analyses in six cortical and subcortical brain regions. Our results show varying levels of network homology between males and females across brain regions, although the associations between these structures and the expression of MDD remain highly sex specific. We refined these associations to several symptom domains and identified transcriptional signatures associated with distinct functional pathways, including GABAergic and glutamatergic neurotransmission, metabolic processes and intracellular signal transduction, across brain regions associated with distinct symptomatic profiles in a sex-specific fashion. In most cases, these associations were specific to males or to females with MDD, although a subset of gene modules associated with common symptomatic features in both sexes were also identified. Together, our findings suggest that the expression of distinct MDD symptom domains associates with sex-specific transcriptional structures across brain regions.

Major depressive disorder (MDD) is a highly pervasive and recurrent disease affecting yearly more than 280 million people worldwide[1]. Notably, the prevalence of MDD is 2 to 3 times higher in women. Women also experience greater symptom severity, with younger age of onset and higher recurrence[2,3]. Moreover, specific syndromes linked to hormonal cycle and pregnancy are also contributing to the sex-specific clinical manifestations of the disease[4]. Despite its major burden on modern societies, current strategies to treat the syndrome remain relatively inefficient with only a third of patients showing complete remission and roughly two third exhibiting various levels of recurrence[5]. These limitations in MDD treatment likely result in large part from its clinical heterogeneity.

[1]CERVO Brain Research Centre, Quebec, QC, Canada. [2]Department of Social and Preventive Medicine, Faculty of Medicine, Université Laval, Québec, QC, Canada. [3]Department of Psychiatry and Neurosciences, Faculty of Medicine, Université Laval, Quebec, QC, Canada. [4]Nash Family Department of Neuroscience and Friedman Brain Institute, Icahn School of Medicine at Mount Sinai, New York, New York, USA. [5]Department of Psychiatry, The University of Texas Southwestern Medical Center, Dallas, TX, USA. [6]McGill Group for Suicide Studies, Douglas Mental Health University Institute, Montreal, Canada. [7]Department of Psychiatry, McGill University, Montreal, Canada. [8]Department of Mathematics and Statistics, Laval University, Québec, QC, Canada. ✉ e-mail: benoit.labonte@fmed.ulaval.ca

Based on clinical symptoms, MDD is a highly heterogeneous syndrome defined by the expression of depressed mood and anhedonia[6]. These two core symptoms are accompanied variably by cognitive impairment, anxiety, weight change, fatigue, agitation, sleep abnormalities, feelings of worthlessness, recurrent thoughts of death, and suicidal ideations, among other symptoms[6]. These symptoms not only vary across patients, but their expressions also evolve in individuals along with the chronicity of the illness. Accordingly, previous attempts to cluster and treat patients based on their symptomatic profiles have shown only modest success[7]. Part of this lack of success can be explained by the way the syndrome is defined and diagnosed. Indeed, as of now, MDD diagnosis is based entirely on standardized, but yet subjective, behavioral measures, and the severity of depressive episodes is obtained by summing up symptoms' presence instead of their intensity[7]. However, these symptoms are broadly different in many dimensions, including their underlying biological substrates and associated molecular mechanisms[8,9].

Findings from functional imaging studies have culminated in the creation of different models linking the expression of specific MDD-relevant clinical features with the activity of distinct brain regions and circuits[10,11]. For instance, hyper-connectivity/activation between the default mode network—Negative Affect Sad and Negative Affect Threat networks—and their respective brain regions, have been associated with the expression of rumination[12], sadness and hopelessness (negative bias)[12,13] and threat dysregulation (scariness and sense of failure)[14,15], respectively. These studies also suggest that the expression of anxiety[16,17], inattention and cognitive dyscontrol (poor concentration, indecisiveness)[7,18–20] and anhedonia[21,22] associated with global hypo-connectivity/activation of the Salience, Attention and Cognitive Control and Positive Affect Happy networks, respectively. These findings support the idea that changes in the activity of specific brain regions and circuits drive the expression of distinct clinical features of MDD, even though the molecular mechanisms underlying these functional changes in each respective brain region remain poorly understood.

Transcriptional changes affecting not only gene expression but also the organization of gene networks have been reported across several brain regions and circuits in post-mortem brain tissues of MDD patients and mouse chronic stress models[23–31]. For instance, detailed analyses revealed the role of gene networks in mediating stress susceptibility in a sex-specific fashion by interfering with intracellular cascades regulating neuronal activity[27,28]. More recently, transcriptional signatures in brains of MDD patients have been associated with trait versus state depression, revealing gene profiles associated with the transition between both clinical states in males[32]. However, none of these studies have been able to show whether any of these transcriptional signatures are associated with the expression of the symptomatic manifestations of MDD in males and females.

Data-driven system-based approaches have shown their advantages over conventional methods in revealing pathogenic etiologies for complex and heterogeneous neuropsychiatric disorders. Network-based analyses, in particular, provide the tools and statistical approaches to classify sub-types of complex diseases according to their molecular profiles. These methods have been used to highlight the molecular architecture underlying the expression of several complex neuropsychiatric disorders such as Alzheimer's disease, autism, bipolar disorder, schizophrenia, and MDD[28,33–35]. Here, we used network analyses to evaluate the potential association between transcriptional signatures across brain regions and the expression of distinct symptom domains relevant to MDD in males and females.

## Results

The main objective of this study is to determine whether transcriptional signatures across different brain regions are associated with the expression of specific symptomatic profiles in males and females with

MDD. To do so, we first mapped transcriptional signatures in males ($n = 25$) and females ($n = 25$) with MDD compared to healthy control subjects (17 males and 22 females) across six brain regions (see Supplementary Table 4 for detailed cohort composition by brain region). We then established the degree of transcriptional changes—both at gene level and network level—across six brain regions in males and females with MDD using differential expression analysis and WGCNA. We explored male and female transcriptional profiles to identify unique and shared associations with the expression of specific symptoms of MDD in both sexes. Finally, we identified hub and node genes for each network and calculated their contribution to the association between their respective module and specific symptoms of MDD in both sexes. Overall, our results show that specific transcriptional signatures are associated with the expression of distinct symptomatic profiles and reveal the molecular substrates underlying the expression of MDD and its clinical manifestations across the brains of males and females.

### Differential expression reveals sex- and brain region-specific transcriptional signatures in MDD

We first used differential expression analysis to identify genes significantly up- or downregulated across six brain regions of males and females with MDD. Our analyses revealed a large number of genes differentially expressed (DEG, $p < 0.05$) in males and females across every brain region with a small proportion of overlap between the two sexes (Fig. 1a; Supplementary Tables 5, 6). In total, we identified between 3.2% up to 35.9% of overlapping DEGs in males and females with MDD across brain regions. The overlap was smaller in the NAc (3.2%) and vSub (4%) between males and females with MDD, while greater overlaps were seen in PFC regions: 35.9% in dlPFC and 11.9% in vmPFC.

To further characterize the transcriptional overlap between males and females with MDD, we used a rank-rank hypergeometric overlap (RRHO) analysis to compare transcriptional signatures from both sexes without restricting our analysis to stringent statistical thresholds[36,37]. Interestingly, our results revealed a significant overlap (max. $P$-value = 1.0E−250) for genes commonly up- or downregulated in both males and females, but only in cortical regions including the OFC, vmPFC dlPFC and to a lower extent the aINS (Fig. 1b). In contrast, RRHO analysis revealed a lack of transcriptional overlap in limbic structures including the NAc and vSub (Fig. 1b). Importantly, our analysis at gene level supports these observations with the most variable genes from our datasets showing sex- and brain region-specific transcriptional changes (Fig. 1c). For instance, *ZNF729*, *RXFP3*, *OR52A5*, *EIF4EBP2* and *CARTPT* exhibit opposite transcriptional patterns between males and females with MDD across brain regions. Furthermore, genes such as *NR4A1*, *HSPAL2B*, *PCDHB4*, *RPS26,* and *SUCNR1* show consistent changes across brain regions while others exhibit region-specific transcriptional regulation.

We then used gene ontology analysis to identify functional features enriched with DEGs across each brain regions. As expected, this analysis identified functional terms broadly different between males and females across each brain region (Supplementary Tables 7, 8). However, we also identified a subset of functional domains shared between males and females with MDD, including GABAergic synaptic function in the aINS, neuropeptide signaling pathway in the vmPFC and AMPA receptor activity and neurotransmitter receptor complex in the vSub (Supplementary Fig. 1). These findings suggest that sex-specific transcriptional changes converge onto some similar functional alterations across brain regions in males and females.

Finally, to confirm the reproducibility of our findings, we overlapped our transcriptional profiles with those from our previous work[31] using DEG and RRHO analyses. Importantly, our examination revealed a strong and consistent overlap between results obtained in this new cohort and those from males and females with MDD previously

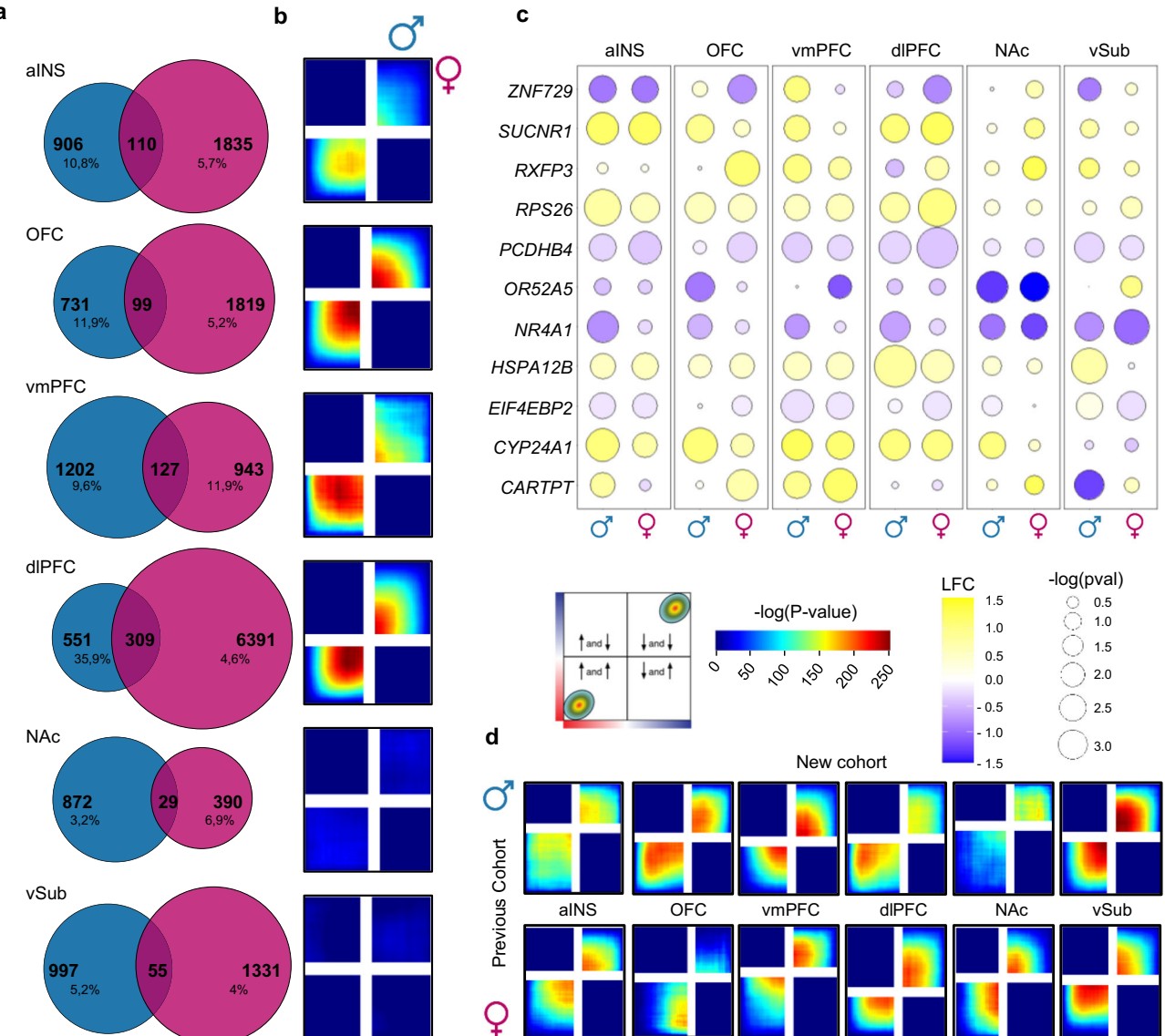

**Fig. 1 | Differential gene expression patterns differ in males and females with MDD, revealing sex- and brain region-specific transcriptional profiles. a** Venn diagrams of DEGs (nominal $p < 0.05$; two-tailed) showing low overlap between male MDD (blue) and female MDD (pink) across brain regions. **b** RRHO maps displaying transcriptional overlaps between male MDD and female MDD across brain regions. Signals in the bottom left and upper right quadrants represent an overlap for commonly upregulated and commonly downregulated genes, respectively. The color bar represents the degree of significant ($-\log$ ($p_{adj}$-value); FDR corrected; two-tailed) overlap between transcriptional signatures in males and females with MDD. **c** Bubble plot showing the most variable genes differentially expressed in male MDD and female MDD across six brain regions. Colors represent log fold change values, with blue for genes downregulated and yellow for genes upregulated in MDD versus control conditions. The radius of the circles shows significance levels according to their $p$-values (nominal $p < 0.05$; two-tailed). **d** RRHO maps representing transcriptional overlaps between the new cohort sequenced in this paper and a previously published cohort[27]. Source data are provided as a Source Data file.

published on the same brain regions tested both at the gene (DEG) and broader (RRHO) signature levels (Supplementary Fig. 2; Fig. 1d). Overall, results from our analyses are consistent with previous comparisons of DEG profiles in males and females with MDD[25,30,31] and suggest that transcriptional signatures in both sexes exhibit broad differences but with a certain level of overlap, mainly in cortical brain regions, ultimately affecting genes converging onto common functional pathways across brain regions in males and females with MDD.

## WGCNA highlights region-specific gene networks associated with MDD in males and females

We next used WGCNA to construct the transcriptional organization of gene networks in males and females with and without MDD across all six brain regions. Importantly, as opposed to previous strategies of pooling brain regions[31], the size of our current combined cohort allowed us to create sex-specific gene modules for all six brain regions investigated. Gene ontology analysis was used to associate a functional term to each of these modules. In total, we identified between 20 to 109 gene modules composed of between 50 to 7662 genes across every brain region representing in an unbiased manner the vast majority of functional domains relevant to brain activity (i.e., synaptic function, metabolic function, cytoskeletal plasticity, immune function, etc.) in males and females with and without MDD (Supplementary Fig. 3; Supplementary Table 9).

We started by testing the extent to which the transcriptional organization of gene networks is preserved across the brain of males and females with MDD. To do so, we calculated $Z_{summary}$ values for modules in the male MDD group and considered every module with a $Z_{summary}$ score higher than 10 as being preserved in the female MDD. Not surprisingly, we found a significant proportion (from 35% to 77%)

of male gene networks preserved in females across all six brain regions (Supplementary Fig. 4). The NAc (76.8%), aINS (69.7%) and vmPFC (66.3%) showed the highest level of preservation in modules found in male versus female MDD, while a smaller but still considerable proportion of gene networks were preserved in the vSub (34.6%), OFC (52.2%) and dlPFC (55.0%). Together, this suggests that similar transcriptional structures are involved in the control of fundamental cellular processes across brain regions in males and females.

We then tested whether these modules, either preserved or unique in both sexes, were associated with changes in intramodular connectivity. Intramodular connectivity represents the strength of the overall correlation values amongst genes from the same module. Change in intramodular connectivity between two conditions (i.e., MDD versus CTRL), defined as module differential connectivity (MDC), has been associated with dysfunctional organization of transcriptional structures in stressed mice and humans with MDD[25,31,38,39]. Interestingly, our analysis suggests that a large proportion of unique modules in males and females with MDD associated with a significant MDC compared to their respective controls. In female MDD, this proportion reaches 67.3% in vSub, 57.9% in the NAc and 53.3% in the OFC, while no unique module in female aINS showed differential connectivity (Fig. 2a, b). Similar, although lower, proportions were found in males with MDD: the vSub (41.4%), NAc (36.8%) and vmPFC (27.9%) showed the highest proportion of modules associated with intramodular changes in structural connectivity compared to their respective controls (Fig. 2a, b). For both males and females, modules with a significant GOC or LOC were enriched for genes relevant for different functional terms, such as transcription factor activity (Male aINS and OFC), BDNF signaling (female dlPFC), neuropeptide activity (female NAc) and synaptic activity (male vSub) in a region-specific fashion (Fig. 2b).

In contrast, a much smaller proportion of modules preserved between males and females with MDD exhibit significant MDC compared to their respective control conditions. Indeed, with the exception of the NAc (31.7%) in male MDD and the vmPFC (21%) in female MDD, the proportion of sex-preserved modules associated with a significant MDC compared to their respective controls was lower than 20% in every brain region (Fig. 2c, d), with the OFC and aINS showing no module associated with MDC in either males or females with MDD. On the contrary, our analysis revealed a large proportion of modules preserved in males and females with MDD associated with a significant MDC when compared to the other sex (Male MDD versus Female MDD). Indeed, this proportion reached 72.7% in the dlPFC, 67.6% in the vSub and 42.9% in the NAc, while lower levels were found in the aINS (8.7%), vmPFC (17.5%) and OFC (20.0%) (Fig. 2c, d). Modules preserved in both sexes associated with a significant GOC or LOC were also enriched for genes relevant for different functional terms, including synaptic function (aINS, vmPFC, dlPFC, NAc, vSub), function of the mitochondria (aINS, OFC, vSub), intracellular protein signaling (aINS, vmPFC, NAc, vSub) and nuclear control of gene expression (aINS, vmPFC) (Fig. 2d). Together, this suggests that a significant proportion of the transcriptional organization of gene networks is shared across the brain of males and females with MDD. Nevertheless, despite this high level of homology, our findings demonstrate that both unique and sexually preserved gene modules contribute to the expression of MDD distinctly in males and females via changes in their structural connectivity and underlying biological functions. Furthermore, brain regions such as NAc and dlPFC are more importantly associated with MDD in both sexes than the aINS.

## Transcriptional associations with symptomatic features in males and females with MDD

Our analyses to this point show that networks of co-expressed genes in males and females contribute to the expression of MDD in both sexes. We next tested whether transcriptional signatures across brain regions in males and females with MDD associated with specific symptomatic profiles in both sexes. Clinical information for each sample was obtained by means of post-mortem psychological autopsies as described before[40,41]. Globally, symptomatic data obtained through this approach provided information on changes in appetite/weight, insomnia/hypersomnia, psychomotor agitation/retardation, low self-esteem and difficulty in concentration/indecision (Supplementary Table 2) with similar proportions in males and females with MDD. We first ran a hierarchical clustering analysis to test whether the expression of specific symptomatic features associated with variations in gene expression across brain regions in males and females. Interestingly, our analysis in females revealed clear patterns of gene expression across all brain regions in samples expressing insomnia/hypersomnia (Fig. 3a). Similar patterns were observed in the aINS, vmPFC, and dlPFC for samples with psychomotor agitation/retardation and for changes in appetite/weight in the vmPFC and vSUB. In contrast, no such clear patterns were identified in males with MDD (Fig. 3a).

We then expanded our analysis to assess whether the integrative and correlative features of WGCNA would provide further advantages in revealing significant associations between male and female MDD symptomatic profiles and transcriptional gene networks. To do this, we constructed gene networks combining transcriptional profiles from males and females with and without MDD (Supplementary Fig. 5) and calculated module eigengene values for samples in every brain region by extracting the first principal component from each identified module. For every given symptomatic feature, we measured the associations between that specific symptom and the module eigengene values using biserial correlations. This approach identified several modules associated with the expression of each symptom of MDD, although these associations differed by sex and brain region (Fig. 3b). For instance, the largest proportion of modules in males associated with change of appetite/weight was found in the OFC (26.9%), psychomotor agitation/retardation in the NAc (31.0%), low self-esteem in the dlPFC (26.1%) and difficulty in concentration/indecision in the OFC (11.5%). In females, the highest proportion of modules associated with change of appetite/weight was found in the dlPFC and aINS (13.0% and 12.7%, respectively), psychomotor agitation/retardation in the dlPFC (30.4%), low self-esteem in the aINS (12.7%) and difficulty in concentration/indecision in the vmPFC (46.0%). Overall, the expression of insomnia/hypersomnia was associated with a larger proportion of gene networks across the brain, in both males and females, compared to other symptoms (Fig. 3b; Supplementary Fig. 6).

We next investigated which of these gene networks are the most strongly associated with the expression of specific symptoms across brain regions of males and females with MDD. For instance, *Ivory* in the OFC (Fig. 4a) is the gene network most strongly associated with the expression of change in appetite/weight in males with MDD ($r = 0.76$; $p_{adj} < 0.001$) (Fig. 4b). Noticeably, this network is also associated with the expression of insomnia/hypersomnia ($r = 0.68$; $p_{adj} < 0.005$) (Fig. 4c) and difficulty in concentration/indecision ($r = 0.73$; $p_{adj} < 0.005$) (Fig. 4d) in male MDD. *Ivory* is enriched for genes associated with synaptic signaling, most importantly GABAergic neurotransmission ($p_{adj} < 5.0^{E-8}$) (Fig. 4f). Alterations of the GABAergic system in the PFC have been frequently associated with the expression of MDD[42,43], although its role in specific symptomatic features has never been reported. Interestingly, *Ivory* is depleted of DEGs but is enriched for genes significantly associated with the expression of each three symptoms in males (Fisher Exact Test (FET): change in appetite/weight, $p_{adj} < 1.0^{E-18}$; insomnia/hypersomnia, $p_{adj} < 5.0^{E-11}$; difficulty in concentration/indecision $p_{adj} < 5.0^{E-17}$) (Fig. 4e), including all three hub genes, namely, GAD1 and GAD2 which encode glutamate decarboxylase 1 and 2 and NXPH1 which encodes neurexophilin 1 (Fig. 4a). In addition to *Ivory* in the OFC, we also identified *Darkviolet* in the NAc and *Darkred* in the dlPFC which are associated with the expression of psychomotor agitation/retardation ($r = -0.69$; $p_{adj} < 0.0001$) and low

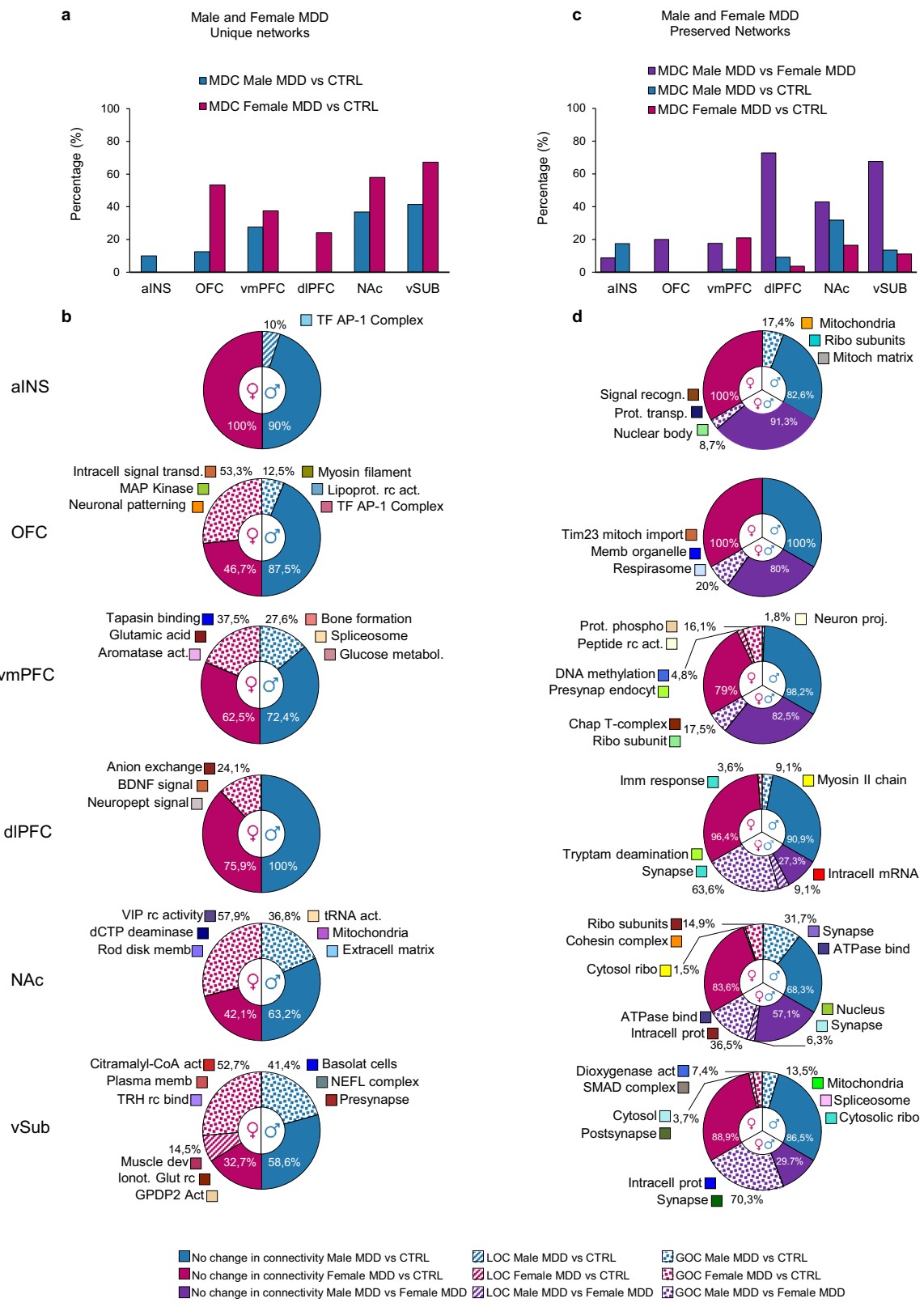

**Fig. 2 | Changes in the structural connectivity of sexually preserved/unique modules contribute to the expression of MDD. a** Proportion of unique modules with significant MDC in MDD versus CTRL in males (blue) and females (pink). **b** Proportion of unique modules with significant LOC (striped area) and GOC (dotted area) and their functional ontological terms. **c** Proportion of preserved modules with significant MDC in MDD versus control in males (blue) and females (pink) and in male MDD versus Female MDD (purple). **d** Proportion of preserved modules with significant LOC (striped area) and GOC (dotted area) and their functional ontological terms. Source data are provided as a Source Data file.

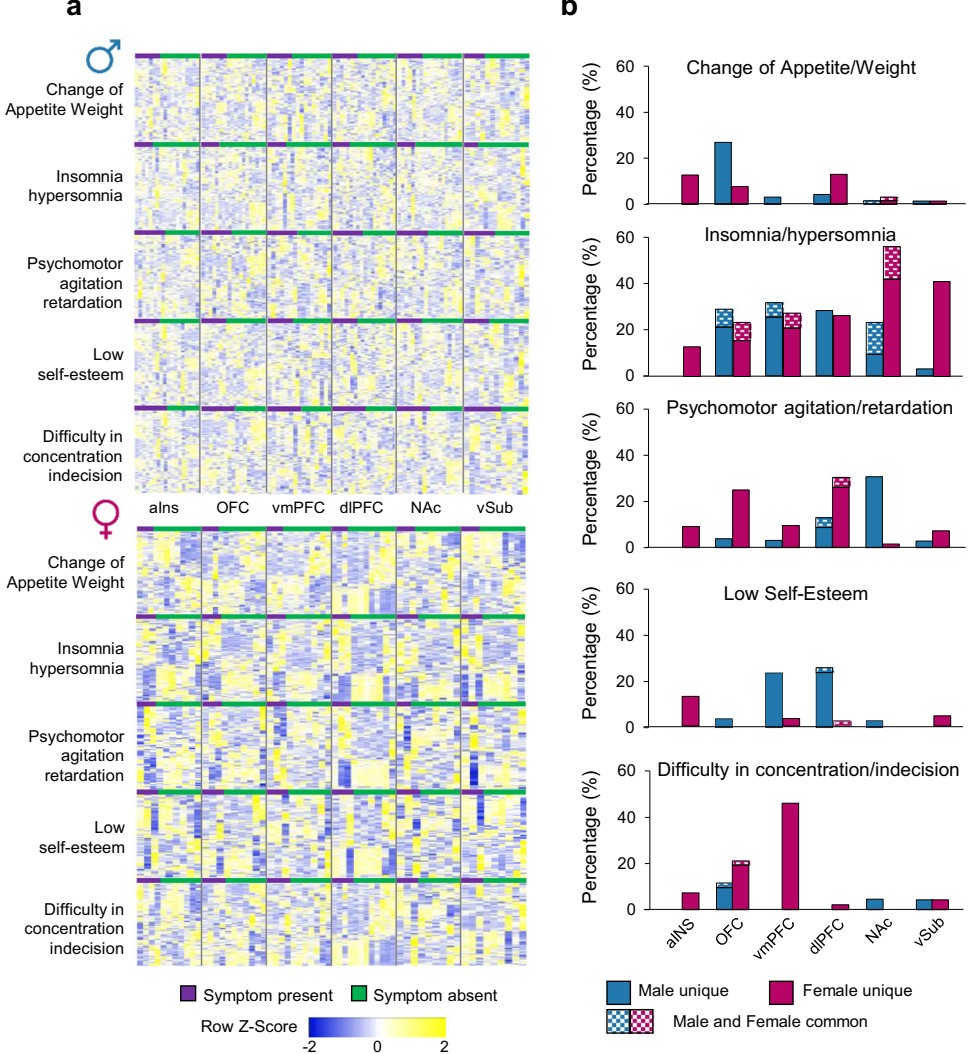

**Fig. 3 | Symptomatic profiles in MDD associated with transcriptional alterations across brain regions in males and females. a** Heatmaps displaying hierarchical clustering of the most variable genes according to the presence (purple) or absence (green) of symptoms. **b** Proportion of gene co-expression modules associated with symptomatic features in males (blue) and females (pink) with MDD. Stippled areas show percentage of modules in males and females associated with the same symptoms in both sexes. Source data are provided as a Source Data file.

self-esteem ($r = -0.83$; $p_{adj} < 0.0001$) in males with MDD, respectively (Supplementary Fig. 7).

In females, our analysis pointed to *Darkorange* in the aINS as a module associated with the expression of change in appetite/weight uniquely in female MDD (Fig. 5a). *Darkorange* is enriched in genes relevant to synapse ($p_{adj} < 1.0^{E-9}$) and cell junction ($p_{adj} < 1.0^{E-8}$). Importantly, *Darkorange* in female MDD is enriched in downregulated DEGs ($p_{adj} < 3.0^{E-6}$) and also associated with the expression of change in appetite/weight ($p_{adj} < 1.0^{E-57}$). In fact, 39% of all genes in *Darkorange* are associated with the expression of this symptom in female MDD ($p_{adj} < 5.0^{E-26}$), including the seven hub genes among which, CLSTN1 and CLSTN3 encoding transmembrane protein calsyntenin family members and PIK4KA encoding phosphatidylinositol (PI) 4-kinase were also significantly downregulated in the aINS of females with MDD (Fig. 5a). As well, we identified *Saddlebrown* in the dlPFC (Fig. 5e) to be significantly associated with the expression of psychomotor agitation/retardation ($r = 0.891$; $p_{adj} < 0.0001$) (Fig. 5f) in female MDD. This module (cellular protein catalytic process; $p_{adj} < 0.0005$) (Fig. 5h) is strongly enriched for upregulated DEGs ($p_{adj} < 5.0^{E-37}$) (Fig. 5f) and with the expression of psychomotor agitation/retardation in the dlPFC of female MDD (34.1%; $p_{adj} < 1.0^{E-17}$) (Fig. 5g), including all 5 top hub genes

*SELENOT*, *ACTR3*, *CHMP2b*, *SGPP1* and *TM9SF3*. Additional modules strongly associated with the expression of insomnia/hypersomnia (*Skyblue*-OFC; $r = -0.747$; $p_{adj} < 0.0001$), low self-esteem (*Purple*-aINS; $r = -0.827$; $p_{adj} < 0.0001$) and difficulty in concentration/indecision (*Palevioletred3*-vmPFC; $r = -0.60$; $p_{adj} < 0.0001$) in females with MDD are highlighted in Supplementary Fig. 8.

Finally, we highlighted a small fraction of gene networks associated with the same symptomatic features in both sexes (Fig. 6a; Supplementary Table 10). In total, we identified 22 gene modules associated with the same symptoms in males and females. However, 17 (77%) showed opposite associations in males and females and divergent genes responsible for these associations. For instance, the expression of the *Orange* module in the OFC (Fig. 6b), enriched in genes involved in synaptic transmission ($p_{adj} < 5.0^{E-9}$), is positively correlated with the expression of difficulty in concentration/indecision in males with MDD ($r = 0.484$, $p_{adj} < 0.05$), but negatively correlated with the same symptom in female MDD ($r = -0.622$, $p_{adj} < 0.05$) (Fig. 6c). Furthermore, out of the genes significantly associated with the expression of this symptom in both sexes (51 in male MDD and 46 in female MDD), only six were common in males and females (Fig. 6d). Noticeably, only one hub gene (*PPP3R1*) was

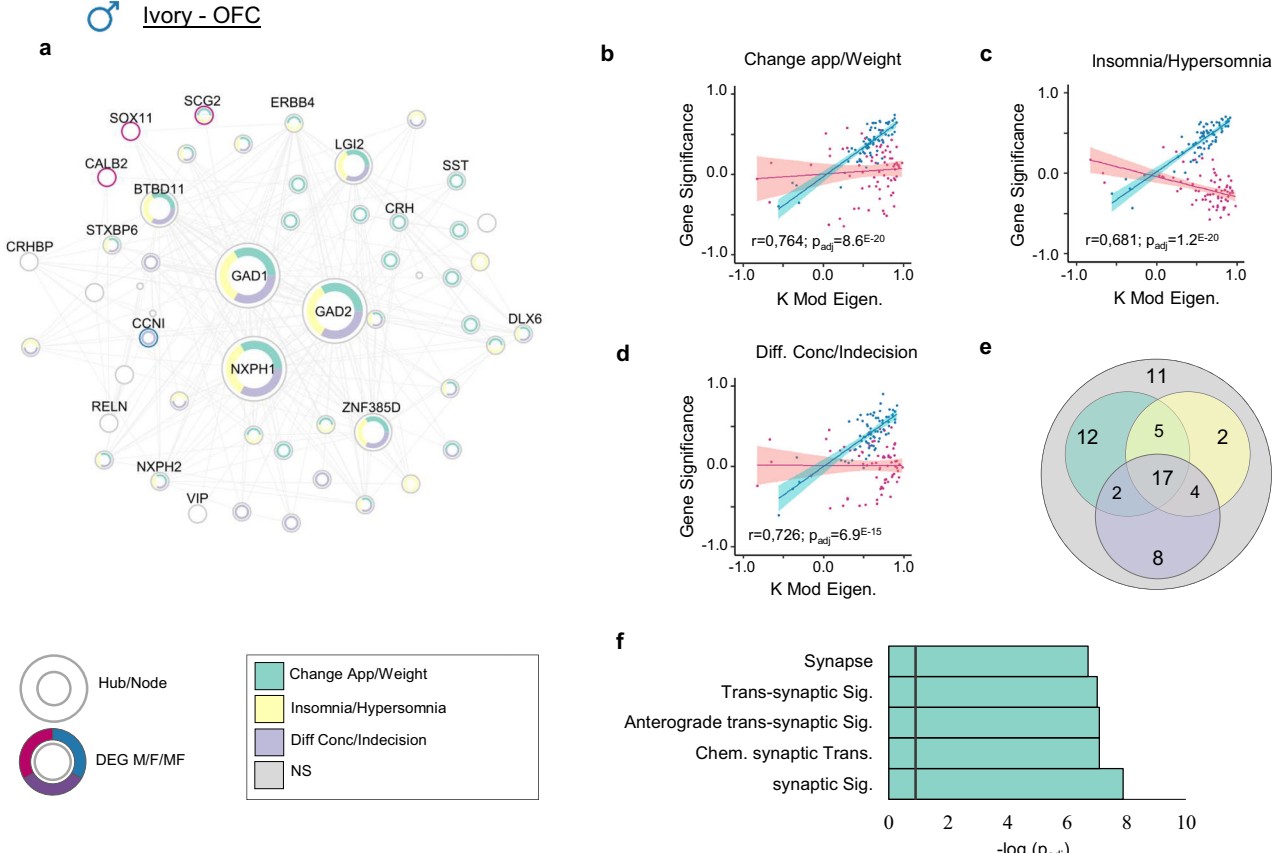

**Fig. 4 | The most significant module (*Ivory*) associated with the expression of change in appetite/weight, insomnia/hypersomnia, and difficulty in concentration/indecision detected in male OFC. a** Hubs and nodes representation of *Ivory* and their associations with clinical symptoms of MDD. The inner colors show to which symptom each gene (hub/node) is associated, with green for change in appetite/weight, yellow for insomnia/hypersomnia, and purple for difficulty in concentration/indecision. The surrounding color shows if the gene is differentially expressed in male (blue), female (pink) or both (purple). Hubs and nodes are defined by the size of the circles. **b–d** Correlation between module membership (KME) and gene significance (GS) values of the association between each gene and relevant symptomatic feature, for genes coexpressed in *Ivory*, in male (blue) and female (pink). Fitted linear regression lines in corresponding colors are employed to demonstrate the linear associations. The 95% confidence band ($\overline{GS} \pm t_{0.975}SE_{\overline{GS}}$) for each fitted regression line is visually indicated with a relevant, brighter color. The higher the correlation, the more relevant the module to the specific symptom. **e** The venn diagram shows the number of genes in *Ivory* module associated with at least one out of the three symptoms, with change in appetite/weight in green, insomnia/hypersomnia in yellow, and difficulty in concentration/indecision in purple. **f** GO enrichment for the *Ivory* module for the first five most significantly enriched GO terms. The black vertical line indicates the significance threshold ($p_{adj}$ <0.05). Source data are provided as a Source Data file.

commonly associated with difficulty in concentration/indecision in males and females with MDD, with all other hub genes being either uniquely associated with this clinical feature in males (*GABRB3*, *PRKCE*) or females (*SYNJ1*, *ATP9A*, *SNAP91*, *RAB6B*) with MDD (Fig. 6b). Similarly, the expression of *Lightcyan1* in the NAc (Fig. 6e), enriched with genes relevant to function of the mitochondria ($p_{adj}$ < 5.0.E-11), was negatively correlated with the expression of insomnia/hypersomnia in male MDD ($r = -0.409$, $p_{adj} = 0.078$), but positively correlated with the same symptom in females with MDD ($r = 0.875$, $p_{adj} < 0.05$; Fig. 6f). Even if we observed associations with opposite directions between males and females, this might be a consequence of combining insomnia and hypersomnia into a comprehensive category labeled sleep disorder. Indeed, previous findings suggest sex-specific prevalence of insomnia and hypersomnia[44].

*Lightcyan1* in NAc is also enriched for genes upregulated in male ($p_{adj}$<0.01) but not female MDD. Six (*GPX4*, *PSMB5*, *PSMB6*, *PRDX5*, *ASNA1*, *EIF4H*) out of seven hub genes were associated with the expression of insomnia/hypersomnia only in females, while *UROD* was the only hub gene common to both sexes (Fig. 6e, g). Together, these findings suggest that while these networks may act as ensembles of genes underlying the expression of clinical features of MDD in both sexes, their specific associations in either males or females may be driven largely by different gene members.

## Discussion

For decades, the expression of MDD's clinical manifestations have been related to variations in the activity of specific brain functions[45–49], while the molecular mechanisms underlying these changes have never been fully explored. Here, we provide a thorough and unbiased description of transcriptional signatures across brain regions associated with the expression of MDD and, more specifically, with the expression of some of its core symptom domains in males and females. Our results suggest that the expression of specific symptoms results from the activity of different gene networks across brain regions. Our findings point toward specific brain structures as being more relevant for the expression of the different symptom domains in males and females. Moreover, although the transcriptional organization of gene networks may be preserved in both sexes, their association with the expression of its symptoms differs significantly in males versus females.

MDD in males and females is defined through the same clinical criteria and both sexes express the same symptoms although at somewhat different levels[50,51]. Accordingly, one may expect similar

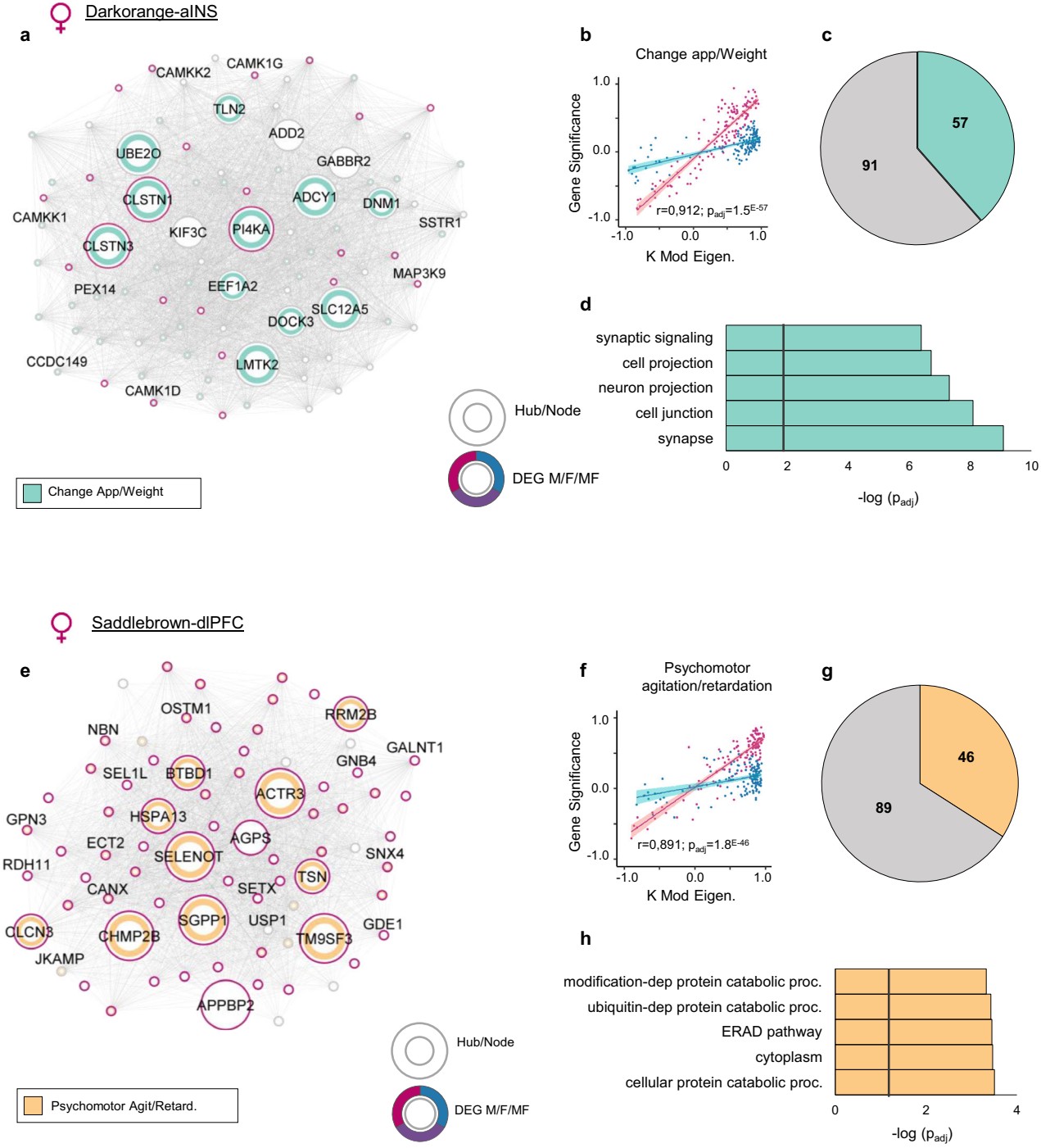

**Fig. 5 | The most significant modules (*Darkorange* and *Saddlebrown*), associated with the expression of change in appetite/weight and psychomotor agitation/retardation in female detected in the aINS and dlPFC respectively. a**, **e** Hubs and nodes representation of (**a**) *Darkorange* and (**e**) *Saddlebrown* their associations with the specific clinical symptoms of MDD. **a** The inner color in green shows the genes significantly associated with the expression of the symptom: change in the appetite/weight. **e** The inner color in orange shows the genes significantly associated with the expression of psychomotor agitation/retardation. The surrounding circle shows if the gene is differentially expressed in male (blue), female (pink) or both (dark violet). Hubs and nodes are defined by the size of the circles. **b**, **f** Correlation between module membership (KME) and gene significance (GS) values of the association between each gene and relevant symptomatic feature, for genes coexpressed in (**b**) *Darkorange* and (**f**) *Saddlebrown*, in male (blue) and female (pink). Fitted linear regression lines in corresponding colors are employed to demonstrate the linear associations. The 95% confidence band ($\overline{GS} \pm t_{0.975}SE_{\overline{GS}}$) for each fitted regression line is visually indicated with a relevant, brighter color. The higher the correlation, the more relevant the module to the specific symptom. **c**, **g** The pie chart shows the number of genes in (**c**) *Darkorange* associated with change in appetite/weight in green and (**g**) in *Saddlebrown* associated with psychomotor agitation/retardation in orange. **d**, **h** GO enrichment for (**d**) *Darkorange* and (**h**) *Saddlebrown* modules for the first five most significantly enriched GO terms. The black vertical line indicates the significance threshold ($p_{adj} < 0.05$). Source data are provided as a Source Data file.

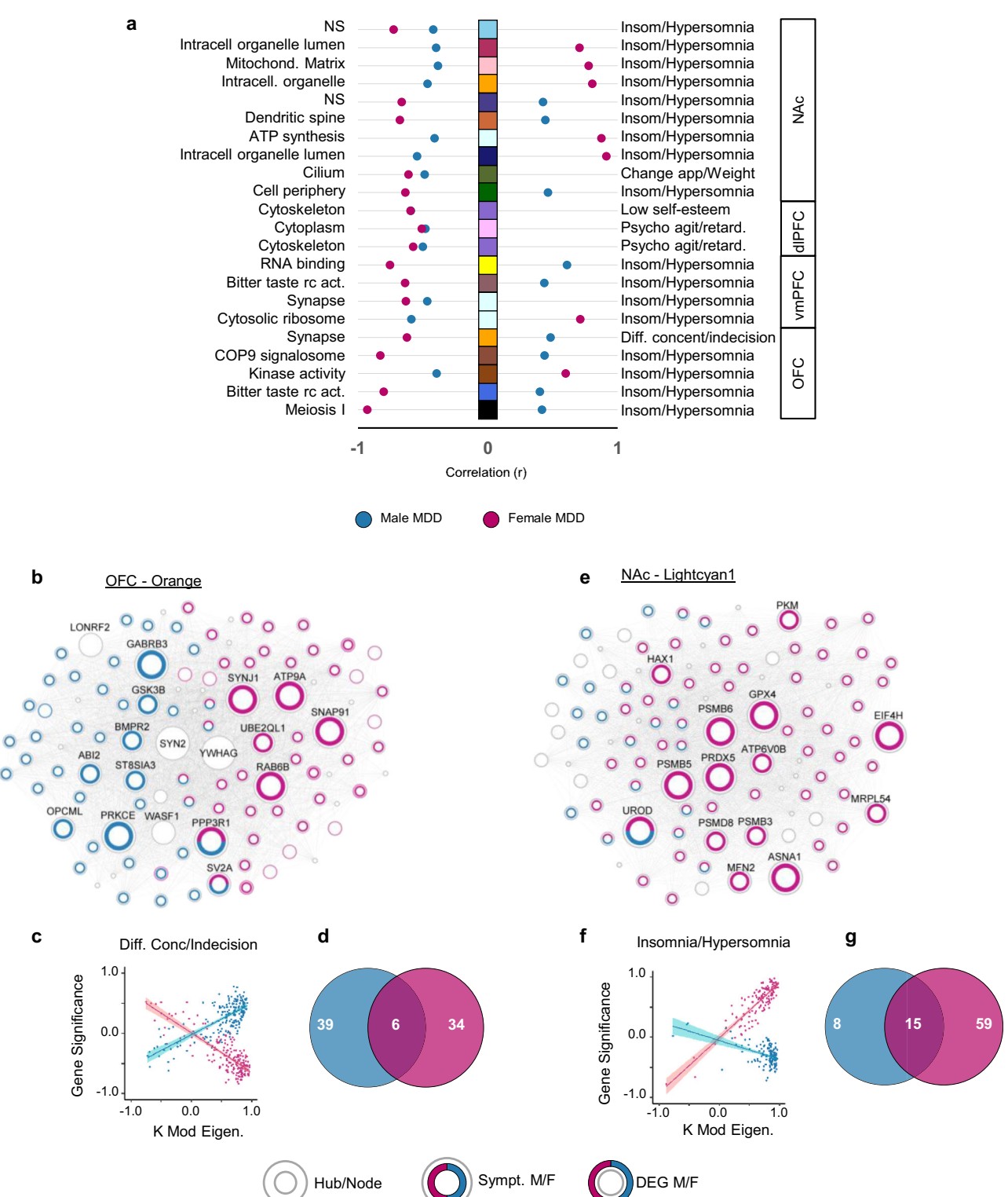

**a** (from top to bottom, left labels): NS, Intracell organelle lumen, Mitochond. Matrix, Intracell. organelle, NS, Dendritic spine, ATP synthesis, Intracell organelle lumen, Cilium, Cell periphery, Cytoskeleton, Cytoplasm, Cytoskeleton, RNA binding, Bitter taste rc act., Synapse, Cytosolic ribosome, Synapse, COP9 signalosome, Kinase activity, Bitter taste rc act., Meiosis I

Right labels (top to bottom): Insom/Hypersomnia, Insom/Hypersomnia, Insom/Hypersomnia, Insom/Hypersomnia, Insom/Hypersomnia, Insom/Hypersomnia, Insom/Hypersomnia, Insom/Hypersomnia, Change app/Weight, Insom/Hypersomnia, Low self-esteem, Psycho agit/retard., Psycho agit/retard., Insom/Hypersomnia, Insom/Hypersomnia, Insom/Hypersomnia, Insom/Hypersomnia, Diff. concent/indecision, Insom/Hypersomnia, Insom/Hypersomnia, Insom/Hypersomnia, Insom/Hypersomnia

Brain regions (right): NAc, dlPFC, vmPFC, OFC

X-axis: Correlation (r) from -1 to 1

Legend: ● Male MDD   ● Female MDD

**b** OFC - Orange

**e** NAc - Lightcyan1

**c** Diff. Conc/Indecision — Gene Significance vs K Mod Eigen.

**d** Venn: 39 | 6 | 34

**f** Insomnia/Hypersomnia — Gene Significance vs K Mod Eigen.

**g** Venn: 8 | 15 | 59

Legend: ◯ Hub/Node   ◯ Sympt. M/F   ◯ DEG M/F

molecular mechanisms underlying the expression of these symptoms in both sexes. However, our results suggest otherwise, even though the transcriptional organization of gene networks is strongly conserved in both sexes. Indeed, we only identified a small proportion of modules associated with the expression of the same symptoms in males and females and, for most of them, we saw opposite correlations between gene networks and symptom in males versus females, consistent with a prior report[22]. For instance, we identified the *Orange* module in the OFC and *Lightcyan1* module in the NAc associated with the expression

of difficulty in concentration/indecision and insomnia/hypersomnia, respectively, in both males and females. These two modules are enriched in genes involved in synaptic transmission and function of the mitochondria, two fundamental processes in males and females that have been implicated previously in MDD[29–32]. However, while this may support the idea of common functional and behavioral implications for these two gene networks in male and female symptomatology, it also suggests that these genes may be acting differently in both sexes. This is further supported by our findings showing that the genes

**Fig. 6 | Common symptoms in males and females are driven in part by the activity of different genes. a** List of gene networks associated with the same symptomatic features in both sexes, with their respective GO terms on the left, and symptoms across brain regions on the right. The squares in the middle show modules with their assigned arbitrary names (colors). The dots show the correlation values between module membership (KME) and gene significance (GS) values, with blue for males and pink for females. **b, e** Hubs and nodes representation of (**b**) module *Orange* in the OFC, showing less than 4% of the genes in this module commonly associated with the expression of difficulty in concentration/indecision in males and females. **e** Module *Lightcyan1* in the NAc in which 10% of the genes are commonly associated with the expression of insomnia/hypersomnia in males and females. **c, f** Correlations between module membership (KME) and gene

significance (GS) values of the association between each gene and relevant symptomatic feature, for genes co-expressed in (**c**) module *Orange*, and (**f**) module *Lightcyan1* in males (blue) and females (pink). Fitted linear regression lines in corresponding colors are employed to demonstrate the linear associations. The 95% confidence band ($\overline{GS} \pm t_{0.975} SE_{\overline{GS}}$) for each fitted regression line is visually indicated with a relevant, brighter color. Opposite direction of this association in males and females reveals an opposite structural association in the two sexes. **d, g** Venn diagrams showing the number of genes in (**d**) module *Orange* associated with difficulty in concentration/indecision and (**g**) module *Lightcyan1* associated with insomnia/hypersomnia in males (blue), females (pink), or in common in both sexes. Source data are provided as a Source Data file.

significantly associated with the expression of these symptoms in males and females with MDD were drastically different. In the OFC, hub genes in the *Orange* module encoding the glycogene synthase kinase 3 (*GSK3B*) and the GABA receptor subunit ß3 (*GABRB3*) were specifically associated with the expression of difficulty in concentration/indecision in male MDD, while hub genes encoding synpatojanin 1 (*SYNJ1*) and the clathrin coat assembly protein AP180 (*SNAP91*) drove the same associations in women. These genes may modulate neurotransmission differently in males versus females in pathological contexts. Overall, while the precise mechanisms underlying these effects remain to be elucidated, our findings further strengthen the hypothesis[31] that common functional pathways are affected in both males and females with MDD, although through the action of different genes, and expand this concept to specific symptomatology of MDD.

Changes in network structures found in specific brain regions may interfere with regional activity and consequently with the functional connectivity of brain networks controlling specific behavioral domains relevant to the expression of each symptom in males and females[10,11]. Here, although we did not empirically confirm the functional roles of our predicted gene networks, converging evidence in mouse models of chronic stress suggests that this may be the case[27,31,38,39]. For instance, the *Ivory* gene network in the OFC, associated with the expression of three main symptoms of MDD in males, is enriched for genes relevant to GABAergic neurotransmission. Alterations of the GABAergic system are a hallmark of MDD and have been associated with disruption of the homeostatic inhibitory control over excitatory tone that is required for the top-down processing of cognitive and emotional information in cortical regions[42,43]. As part of a larger brain network, the OFC is hypothesized to be part of the attention and cognitive control circuitry, with alterations of this circuit causing indecision and decreased concentration and attention[7,18–20]. Similarly in females, we identified gene networks associated with the expression of change in appetite/weight (*Darkorange*) and low self-esteem (*Purple*) in the aINS, psychomotor agitation/retardation (*Saddlebrown*) in the dlPFC and difficulty in concentration/indecision (*Palevioletred3*) in the vmPFC that could also associate with changes in the activity of each of these brain regions. It is likely that changes in these sex-specific gene networks underly symptom expression in males or females by interfering with the activity of brain networks, as was recently shown for a cortical-subcortical circuit during adolescent development[52]. Thus, we hypothesize that the reorganization of precise transcriptional structures across brain regions in males and females may underly the expression of distinct clinical features of MDD in a sex-specific fashion.

Our approach in this study was to break down the complexity of MDD to its simplest expression, i.e., at the symptom level. However, various symptoms often co-occur. This is well exemplified through the clustering approaches we used at the gene level but could not be accounted for at the gene network level. Indeed, although we analyzed one of the largest available RNA-seq datasets for MDD in males and females, we did not reach sufficient power to evaluate the co-occurrence of specific symptoms nor to test association with clusters of symptoms in males and females at the network level. Similarly,

the postmortem nature of our study involved additional limitations such as the incapacity to evaluate the intensity and recurrence of each symptom in both sexes and to assess more precisely each symptom domains. Clinical manifestations such as hypersomnia or insomnia, gain or loss of appetite and psychomotor agitation or retardation are likely to result from different molecular and cellular substrates. While our findings highlight several transcriptional programs associated with each of broader symptom domains, future studies should consider differentiating the molecular substrate underlying more precise clinical manifestations. Nevertheless, we believe adopting a system biology approach, integrating different levels of analysis, provides a much more reliable interpretation of the molecular mechanisms underlying MDD and its clinical manifestations in males and females. Furthermore, our analyses confirmed the reproducibility of our transcriptional observations, further supporting the validity of our findings.

To conclude, the heterogeneity of MDD has long been a brake toward an understanding of its molecular etiologies. Progress in computational biology combined with improvement in the collection of clinical information allowed us to transition from global transcriptional screenings[23–26] to state versus trait transcriptional assessment[32] and sex-specific transcriptional structures[29–31] and ultimately toward the dissection of transcriptional signatures associated with symptomatic profiles in males and females. Findings from this study suggest such associations exist and strongly support the implementation of systems biology approaches to larger longitudinal cohorts with evolving pathological states. Furthermore, converging transcriptional signatures have been identified across several psychiatric conditions[30,53]. Dissecting the transcriptional signatures underlying the expression of clinical manifestations either common or unique to these conditions will further improve our capacity to diagnose each condition with greater precision. Indeed, combining data from large-scale GWAS studies with the approaches described here on peripheral tissue could provide new tools to stratify patients. Ultimately, this would improve our understanding of the molecular mechanisms underlying the expression of these conditions leading toward personalized ways to diagnose and treat symptoms and disease states rather than broad syndromes.

## Methods
Brain tissues were obtained from the Douglas Bell Canada Brain Bank (Douglas Mental Health Institute, Verdun, Québec) and from the University of Texas Southwestern Medical Center Brain Bank. In total, analyses were performed on 89 samples including 25 male MDD, 25 female MDD, 17 male CTRL (healthy controls) and 22 female CTRL. Sociodemographic and clinical information including sex, phenotype (MDD, CTRL), age, pH, postmortem interval (PMI), treatment history, smoking history, history of early life adversity, cause of death, presence of drug and/or alcohol abuse and cohort (Montreal, Texas) are listed in Supplementary Table 1. All analyzes were performed on six brain regions including the anterior insula (aINS), orbitofrontal cortex (OFC; BA11), cingulate gyrus 25 (BA25; cg25; vmPFC [ventromedial prefrontal cortex]), dorsolateral PFC (BA8/9; dlPFC), nucleus

accumbens (NAc) and ventral subiculum (vSub). Overall, we sequenced RNA from 41 new human brains and combined this dataset with RNA profiles from 48 brains published by us before[31]. Postmortem tissues from all six brain regions were carefully dissected at 4 °C after having been flash-frozen in isopentane at −80 °C. All dissections were performed by histopathologists using reference neuroanatomical maps[54].

Psychiatric history and socio-demographic information were obtained via psychological autopsies carried out by trained clinicians using the same methods in case and control groups[40,41]. Diagnosis and clinical information including symptomatic profiles were obtained using DSM-IV criteria by means of SCID-I interview adapted for psychological autopsies[6,55]. Nine main categories of symptoms were recorded, including depressed mood, loss of interest or pleasure, change in appetite/weight, insomnia/hypersomnia, psychomotor agitation/retardation, fatigue or loss of energy, low self-esteem, difficulty in concentration/indecision and recurrent suicidal thoughts. Notably, since depressed mood, anhedonia, fatigue and recurrent suicidal thoughts were expressed by every MDD patient, we did not include those symptoms in our analysis (Supplementary Table 2). The study was approved by the research ethics boards of McGill University and UT Southwestern. Written informed consent was obtained from all participants.

### RNA sequencing

RNA from human postmortem brain samples was extracted using the RNeasy micro kit with Trizol, followed by DNase I treatment, as described by the manufacturer (Qiagen). RNA integrity (RIN) and concentration was quantified using a Bioanalyzer (Agilent). RNA libraries were synthesized from 1 μg of RNA using the ScriptSeq Complete Gold Kit (Epicentre, Illumina) including an initial ribosomal RNA depletion step. Each library was spiked with an external RNA sample as a control as suggested by the manufacturer (Thermofisher). Samples were barcoded and sequenced in multiplex (8 per lane) twice at a depth of 50 million reads (50 bp paired-end) per sample on Illumina HiSeq2500.

### Data processing

Sequencing data from all 89 samples for all 6 brain regions were analyzed using the same criteria. Sequencing quality and trim reads were assessed using FASTQ and FASTX-toolkit. TopHat was used to align paired-end reads to the GENCODE 2019 (GRCh38.p12) human annotation. Overall, every sample included in this study passed QC assessment. Reads for every sample were counted using HTSeq. A gene was considered the union of all its exons in any known isoforms, based on GENCODE annotation. Any reads that fell in multiple genes were excluded from the analysis. Threshold for filtering out genes expressed at low levels was set to <5 reads in at least 20% of the samples per group as described previously[31].

We adapted multiple preprocessing steps to ensure both statistical and biological relevance. Gene expression was first transformed and normalized using the voom function in the limma package[56,57]. Batch effect and potential unwanted sources of variance in gene expression across all samples were identified through RUVseq using spike-in controls[58]. This method is designed to identify any sources of unwanted variation, including the probable heterogeneity between two merged datasets. As expected, the effect of batch was found to be significant for every brain region. The top first factor was extracted and included as a covariate in the differential expression analysis. We then performed a principal component analysis (PCA) to reveal the effect of clinical and technical covariates on variations of gene expression. We identified significant effects for PMI, pH, cohort, drug abuse and RIN in the aINS; age, PMI, pH, childhood abuse, cohort, drug abuse and RIN in the OFC; age, pH, cohort, drug abuse and RIN in the vmPFC; age, PMI, pH, cohort, drug abuse and RIN in the dlPFC; age, PMI, childhood

abuse, cohort, drug abuse and smoking in the NAc; and PMI, cohort, drug abuse and RIN in the vSub (Supplementary Table 3). The effects of these covariates were adjusted in our downstream differential expression and gene co-expression network analyses.

**Differential expression analysis.** Differentially expressed genes (DEGs) were identified through a generalized linear model (GLM) implemented in limma with sex (male and female) and phenotype (MDD and CTRL) as main factors. A single GLM was performed for each brain region controlling for every region-specific covariate identified through our PCA and RUV analyses. An individual gene was called differentially expressed if the nominal $P$-value of its $t$-statistic was ≤0.05. Globally, this approach allowed the identification of genes differentially expressed in males and females with MDD while controlling for baseline variations in gene expression along with the effects of clinical and technical covariates across every brain region.

**Transcriptional overlap analysis.** We used a rank-rank hypergeometric overlap (RRHO) analysis[36,37] to measure transcriptional overlap between males and females with MDD. RRHO was also used to confirm the reproducibility of our results by overlapping results from the current analysis with previously published transcriptional maps in males and females with MDD[28]. Gene lists were ranked and signed according to their degree of differential expression in male MDD and female MDD versus CTRL, respectively ($-\log_{10}(P\text{-value})$) multiplied by the sign of the $t$-statistic. For each comparison, a matrix of hypergeometric $P$-values was created as a result of the iterative statistical tests evaluating the proportion of ranked genes differing from one condition to another. Multiple testing correction was performed using the Benjamini and Yekutieli (BY) method[59]. Adjusted $P$-values were finally heat-mapped with each pixel representing the adjusted $-\log_{10}$ hypergeometric $P$-values of the transcriptional overlap between males and females with MDD.

**Gene ontology analysis.** Gene ontology analysis was performed using g:Profiler2[60] on DEG lists from males and females with MDD across all brain regions with significant enrichment fixed at FDR < 0.05.

**Gene co-expression network analysis.** Weighted gene co-expression network analysis (WGCNA)[61] was used as a systems biology approach to identify modules of highly co-expressed genes. Multiple iterations of WGCNA were performed according to our objectives including (1) for males and females with and without MDD for every brain region (24 networks) and (2) combining males and females with and without MDD for every brain region (6 networks). Five samples were detected as outliers before running network analysis. These outliers were identified via the construction of an Euclidean distance-based sample network with standardized connectivity <−3.5 as the exclusion connectivity threshold. These outliers were removed from the final network construction. Network construction was adjusted for the same covariates used in the differential expression analysis (Supplementary Table 3). Weighted gene co-expression networks were built with a matrix of biweight midcorrelation between all gene pairs which was converted to an unsigned adjacency matrix using a soft threshold power and then transformed into a topological overlap matrix for modular structure detection[62,63]. Highly co-regulated genes were identified through average linkage hierarchical clustering to create groups of genes, with a subsequent dynamic tree cut to explore clusters in a nested dendrogram, with each branch corresponding to a module. Each module was named by a unique arbitrary color and associated with an ontological term using g:Profiler2 from Bioconductor. FDR $p$-values and fold enrichment for each module were reported. Genes with the highest intramodular connectivity (top 5% highest intramodular connectivity) were considered as hub genes. Network organization was represented through Cytoscape v3.9.1[64,65].

Modular enrichment for DEGs was assessed using the GeneOverlap package[66] from Bioconductor. Enrichment was tested for genes significantly upregulated or downregulated in each module in males and females across all 6 brain regions. Fisher's exact tests were used to perform enrichment assessment with significance fixed at $p_{adj} < 0.05$.

**Module differential connectivity.** We used module differential connectivity (MDC) to quantify differences in co-expression network organization between male MDD and female MDD compared to controls and male MDD versus female MDD across brain regions. MDC is determined by calculating the ratio of connectivity between all gene pairs in a module in one condition (phenotype, sex, or brain region) to that of the same gene pair in another condition. MDC values larger than 1 indicate gain of connectivity (GOC) or stronger co-expression between genes, while values lower than 1 indicate loss of connectivity (LOC) or weaker co-expression between genes. The statistical significance of MDC was adjusted for multiple testing using the FDR permutation method[67].

**Module preservation.** Module preservation was carried out to assess whether gene modules in males and females were preserved in the opposite sex, respectively and across brain regions in males and females. Module preservation was computed with the preservation statistics of the WGCNA package. Network preservation statistics do not require independent module identification in a test group. The approach evaluates the preservation of connectivity patterns of the member genes and the distinctiveness of a module as a whole from other modules. Module preservation can be established by four complementary statistics including median rank, $Z_{density}$, $Z_{connectivity}$, and $Z_{summary}$. $Z_{density}$ and $Z_{connectivity}$ statistics are the standardized preservation statistics for density and connectivity, respectively, while $Z_{summary}$ is the average of $Z_{density}$ and $Z_{connectivity}$. Preservation in this study was established through the $Z_{summary}$ measures. Modules with a $Z_{summary}$ score higher than 10 were considered preserved, as recommended[68].

**Clinical association with gene expression and transcriptional modular structure.** Association between gene expression and clinical symptoms across brain regions was calculated by means of hierarchical clustering with complete linkage, using the Pearson correlation coefficient, on the 200 strongest DEGs in each brain region. Heatmaps were made using CPM for the top 200 DEGs versus samples with and without the expression of each symptom in males and females separately.

Symptom association with network structure was performed by calculating point-biserial correlation coefficients between clinical symptoms (dichotomous variable) and module eigengene values (continuous variable). Module eigengene values are defined as the first principal component of each module. It represents global variance within each module and is calculated using the *moduleEigengenes* function in WGNCA[62]. *P*-values were adjusted for multiple testing using a permutation test (1000 permutations). Finally, the relationship between gene membership and symptomatic expression was assessed by means of the correlation between the gene significance (GS) and module membership. This allowed identifying whether specific genes in each module contribute to the module's association with the clinical symptoms. Module enrichment for genes associated with each symptom was measured via Fisher's exact test with *P*-values corrected for multiple testing using Benjamini−Hochberg[69].

**Statistical analysis.** Although sample size calculation was not performed, the sample size in this study is justified based on several previously published reports using similar or even smaller sample sizes and showing the power to detect significant statistical differences. RNA-seq gene expression data for differential expression was

normalized. In total, 89 samples including 25 male MDD, 25 female MDD, 17 male CTRL and 22 female CTRL, from six brain regions (total 534 samples) were included in this study. Overall, transcriptional profiles were generated for 41samples in six brain regions and combined with RNA profiles from 48 samples published by us before[31]. Differential expression was not corrected for multiple testing. RRHO analysis and FET were corrected using the Benjamini−Hochberg method. Network analysis included network construction, module differential connectivity, GO enrichment, module preservation and module differential expression enrichment, were corrected for multiple testing. Associations between clinical symptoms and modular structures in males and females with MDD were calculated with point-biserial correlations and correlations between the gene significance (GS) and module membership. *P*-values calculated for these coefficients were adjusted using permutations and Benjamini−Hochberg adjustment. Details of each analysis are provided above in each respective section.

### Reporting summary
Further information on research design is available in the Nature Portfolio Reporting Summary linked to this article.

## Data availability
Sequencing data and source files have been used in this study to generate all the results. The first cohort is available on NCBI GEO website (accession codes GSE102556). The second cohort will be made available with no restriction on NCBI GEO website upon publication. Any additional data supporting the findings of this study are available from the corresponding author upon request. Source data are provided with this paper.

## Code availability
Major statistical tools, mentioned in more detail in the method section, including WGCNA, limma, RRHO2 are available as R packages at CRAN (http://cran.r-project.org/). We have also provided a more consolidated resource code (as supplementary) with synthetic data for those who are interested in running the codes step by step.

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

## Acknowledgements

B.L., holds a Sentinelle Nord Research Chair, is supported by the Canadian Institutes of Health Research (Grant Nos. 202010PJT-451728 and 202010PJT-451858), and the Natural Science and Engineering Research Council of Canada (Grant No. RGPIN-2019-06496) and Fonds de Recherche en Santé du Québec (FRQS) Junior-2 salary support. This work was also supported by the U.S. National Institute of Mental Health (Grant No. R01MH129306 to EJN).

## Author contributions

S.M. and B.L. conceived the project, designed the experiments and analyses, and wrote the manuscript. B.L. generated the data. T.H.C. contributed to the design and to the analyses. S.M., A.M.P., and A.M.R. analyzed all the data. E.M.P., C.A.T., G.T., and E.J.N. contributed to the data collection. G.T and E.J.N. contributed to the study design. All authors contributed to the preparation of the manuscript.

## Competing interests

The authors declare no competing interests.
