## [Peer Review File · Nature Communications]

Transcriptional dissection of symptomatic profiles across the brain of men and women with depressionREVIEWER COMMENTS

Reviewer #1 (Remarks to the Author):

This manuscript by Mansouri and colleagues seeks to examine changes in transcriptional regulation across the brain in post mortem samples from patients with major depressive disorder. This is a very important question given the overall burden of depression and lack of efficacy of many available treatments for many patients. The work is performed by a leading group in the field and is done with a high degree of rigor.

In my work as a peer reviewer I think it is important to be critical when necessary, but also equally important to not be critically when it is not necessary. This is the latter case. This is important work that is done well and well presented. The bioinformatics are done to the highest standard, and the authors do an admirable job of making this large and complex dataset be as accessible as possible. I think that this manuscript and the accompanying data (which I assume will be made publicly available) will serve as an important resource for the field. I have very minimal comments below. Well done.

- One thing that I think would be worthwhile to add to the Intro/Discussion is the importance of sex differences in clinical depression. It is more prevalent in women, there are specific syndromes linked to the menstrual cycle, pregnancy, and postpartum. I think that this is really some of the impact of the paper and it could be played up a bit more.
- For Figure S3, are the GO terms just the number of genes within that term that also map to the module? Typically a p value for enrichment would also be included. This might be worthwhile to add.

Minor/text changes

- Abstract line 39: Would remove "in males and females" it makes the sentence awkward and is really not necessary.

Reviewer #2 (Remarks to the Author):

The authors present an interesting study. It aims to explore gene expression differences in several brain regions between MDD post-mortem brain samples and controls both in male and female. They used several strategies of analyses (Differential expression genes identification, RRHO, WGNCA). They also explored gene expression differences at the symptom level. Although original and describing a large amount of new and interesting results based on a large set of data, the study suffers from several significant methodological limitations that should be addressed before publication.

They conducted a large set of new analyses but also included past RNA-seq data from a past study. It is

not clear how authors took account potential confounding factors associated with the use of two different set of data.

Many of major depression brain sample could be also associated with death by suicide. Information about cause of death is not provide (or I did not find it, even in table S1) and could be an important confounding factor, in particular in the context of gender differences in suicide prevalence, much higher in male.

It is not clear why DEG analysis, RRHO and then WGCNA analytic strategies are needed to achieve sufficient argument to reach a significant conclusion and contribution of new knowledge, although all these analyses are interesting per se.

I would also suggest to compare male and female to identify potential confounding factor that could explain differences in gene expression profile (e.g. age, clinical history, toxicology at death...).

To study symptom profile is a very interesting approach and a significant novelty in the field. However, I would suggest to include a comparison in symptom profile by gender first as we already know that men and women could differ in clinical presentation.

If symptom profile differs, it would explain transcriptomic differences and this hypothesis should be tested, after controlling for other confounding factors such as death by suicide.

If symptom profile does not differ, then it remains hard to interpret differences at the transcriptional level for specific symptoms.

Moreover, I would suggest to study symptoms profile using more “naturalistic” approaches. Some symptoms are very correlated and it has been described networks of symptoms. These networks could differ across gender too. For sample size issue, it is probably not possible to study network of symptoms here, but I would suggest to study cluster of symptoms rather than isolated symptoms.

Another limitation deserves also more comments, if specific analysis could not be done: symptoms analysis includes symptoms like “sleep disorder” (ie hypersomnia/insomnia) that describe very different symptoms that could be associated with different biological processes. There is no specific reason that insomnia and hypersomnia are supported by the same biological abnormalities. As a consequence, results described in the manuscript should be interpreted with caution.

Please revise the gene name overall the manuscript, in human it is use to be in capital letter

Reviewer #3 (Remarks to the Author):

In this elegant study by Mansouri and colleagues, the authors use gene expression data from six cortical and subcortical regions of the human brain to identify associations between transcriptional profiles and

MDD symptoms in males and females. Using Weighted Gene Co-expression Network Analysis (WGCNA) on RNA sequencing data from human postmortem brain tissue, the authors identify brain region specific gene networks that associate with MDD in males and females. In a follow-up analysis, they also identify gene modules that associate with specific MDD symptoms based on biserial correlations between the module eigengene and symptom expression. Using this sophisticated analysis approach, the study identifies common and sex-specific gene expression patterns and networks across six brain regions that associate with the expression of specific MDD symptoms. The data presented in this paper are highly relevant for our understanding of molecular mechanisms that could potentially be underlying neuropathology in MDD patients and provide new leads for future genetic studies in humans or functional follow-up studies in rodent models. I only have a few comments/requests for clarification.

1. More than half of women with MDD included in this study had a report of childhood abuse (13/25 MDD women vs 3/22 CTRL women). It would be interesting to know how much of the gene expression patterns are driven by a history of childhood abuse, and how the higher levels of childhood abuse in the MDD women may be related to some of the reported sex differences (also taking into account that reports of childhood abuse were largely the same between CTRL males and MDD males).
2. This reviewer could not find the cause of death in the supplementary tables. Did MDD subjects included in this study die of suicide or natural causes or accidents?
3. Supplementary Table 1 shows that, as expected, a majority of MDD females was medicated. However, it is not clear which medications were taken by the subjects, if subjects were medicated at the time of death, and how medication could have influenced the gene expression findings of this study.
4. The final sentence of the Discussion suggests that “Dissecting the transcriptional signatures underlying the expression of clinical manifestations [...] will further improve our capacity to diagnose each condition with greater precision [...]”. This statement should be further elaborated on, because based on just this sentence it is difficult to imagine how gene expression data from post-mortem brain studies can inform diagnosis or treatment of live patients in the future. Could these transcriptional data e.g. in conjunction with large-scale GWAS studies help us better understand pathology or patient stratification in the future?

Reviewer #1 (Remarks to the Author):

This manuscript by Mansouri and colleagues seeks to examine changes in transcriptional regulation across the brain in post-mortem samples from patients with major depressive disorder. This is a very important question given the overall burden of depression and lack of efficacy of many available treatments for many patients. The work is performed by a leading group in the field and is done with a high degree of rigor.

In my work as a peer reviewer, I think it is important to be critical when necessary, but also equally important to not be critically when it is not necessary. This is the latter case. This is important work that is done well and well presented. The bioinformatics are done to the highest standard, and the authors do an admirable job of making this large and complex dataset be as accessible as possible. I think that this manuscript and the accompanying data (which I assume will be made publicly available) will serve as an important resource for the field. I have very minimal comments below. Well done.

1. One thing that I think would be worthwhile to add to the Intro/Discussion is the importance of sex differences in clinical depression. It is more prevalent in women, there are specific syndromes linked to the menstrual cycle, pregnancy, and postpartum. I think that this is really some of the impact of the paper and it could be played up a bit more.

We agree with the reviewer's point. As suggested, we added additional information in the first paragraph of the introduction.

2. For Figure S3, are the GO terms just the number of genes within that term that also map to the module? Typically, a p value for enrichment would also be included. This might be worthwhile to add.

The numbers presented in Figure S3 are the numbers of genes in each specific module. We changed the figure legend to clarify this point. We also added relevant p-values associated with each GO term enrichment as recommended.

3. Abstract line 39: Would remove "in males and females" it makes the sentence awkward and is really not necessary.

We made the recommended change in the abstract.

Reviewer #2 (Remarks to the Author):

The authors present an interesting study. It aims to explore gene expression differences in several brain regions between MDD post-mortem brain samples and controls both in male and female. They used several strategies of analyses (Differential expression genes identification, RRHO, WGNCA). They also explored gene expression differences at the symptom level. Although original and describing a large amount of new and interesting results based on a large set of data, the study suffers from several significant methodological limitations that should be addressed before publication.

1. They conducted a large set of new analyses but also included past RNA-seq data from a past study. It is not clear how authors took account potential confounding factors associated with the use of two different set of data.

As the reviewer mentions, we did perform detailed analyses combining previously published and new datasets. By doing so, we considered two main sources of variability: one referring to the heterogeneity associated with samples coming from two different brain banks (Douglas Bell Canada Brain Bank and the Southwestern Medical Center Brain Bank) and the second referring to the heterogeneity associated with merging 2 different datasets. The first source of heterogeneity was accounted for by defining a “cohort” variable which was included in our principal component analysis and integrated as a covariate into our statistical models. We also used RUVseq to quantify any source of unwanted variance, including batch effect (second source of variance), and included this effect as a covariate in our statistical models accordingly. We believe this approach did correct the potential confounding effect of merging to datasets. This is further explained in the material and method section at line 111-113, 117, and 152-164.

2. Many of major depression brain sample could be also associated with death by suicide. Information about cause of death is not provide (or I did not find it, even in table S1) and could be an important confounding factor, in particular in the context of gender differences in suicide prevalence, much higher in male.

This is an important point. We performed additional analyses to look at the distribution of suicide cases amongst males and females and show that death by suicide is distributed evenly between sexes (FET, P-value>0.99) in our MDD samples (22 out of 25 in male MDD and 22 out of 25 in female MDD). As such, we believe we can rule out this possibility. We also want to stress out the fact that none of the control samples in this study included suicide cases. Consequently, controlling for the “cause of death” factor would have removed a great proportion of the variance associated with the main effect of phenotype (MDD or control). We now provide information on cause of death in Table S1 to allow the readers to better interpret the results.

Distribution of cause of death across sexes (Fisher Exact test)

	Sex		P-value	
	Category	Male		Female
Cause of Death	Suicide	22 (88%)	22 (88%)	>0.99
	Other	3 (12%)	3 (12%)	

3. It is not clear why DEG analysis, RRHO and then WGCNA analytic strategies are needed to achieve sufficient argument to reach a significant conclusion and contribution of new knowledge, although all these analyses are interesting per se.

We believe complex diseases such as MDD cannot be explained by the alteration of unique genes and pathways. Taken individually, each layer of analysis can provide information on the complexity of MDD in males and females and we feel that reporting top differentially expressed genes (DEG), broad transcriptional overlaps, gene ontology categories enriched in DEGs and gene networks is relevant to the scientific community. This being said, we believe adopting a system biology approach, integrating each of these results provides converging evidence for strong associations between the clinical manifestations

of the MDD and gene signatures in males and females. We think this sum of converging evidence cumulated from several levels of analysis is one of the main strengths of our approach.

- I would also suggest comparing males and females to identify potential confounding factors that could explain differences in gene expression profiles (e.g. age, clinical history, toxicology at death...).

Correcting for the effect of potential confounding factors is primordial in the study of human post-mortem data. In this study, we used principal component analyses to evaluate the potential contribution of several covariates on gene expression (details provided in Table S3). In order to account for these effects in males and females with and without MDD, both sexes and phenotypes need to be included in these analyses. Indeed, after defining the effects of these variables on overall gene expression, we implemented generalized linear models and formulized contrasts to determine the effect of MDD in males and females in all six brain regions while controlling for the effects of unwanted covariates. This model requires to include males and females with and without MDD in the same analysis. To our knowledge, our approach respects the recommended guidelines for identifying and correcting for the effects of covariates on gene expression using RNAseq datasets. This procedure is detailed in the method section at lines 152-172.

- To study symptom profile is a very interesting approach and a significant novelty in the field. However, I would suggest to include a comparison in symptom profile by gender first as we already know that men and women could differ in clinical presentation. If symptom profile differs, it would explain transcriptomic differences and this hypothesis should be tested, after controlling for other confounding factors such as death by suicide. If symptom profile does not differ, then it remains hard to interpret differences at the transcriptional level for specific symptoms.

This is an interesting point. We rationalized that equal distribution of each symptom in males and females is important to avoid having potential confounding effects due to unbalanced sample representation. Below, we used Fisher Exact tests to confirm that there is no significant difference in the representation of each symptom in males and females with MDD.

Distribution of symptomatic profiles of depression in males and females (Fisher Exact test)

Clinical Symptoms	Category	Sex		P-value
		Male	Female	
change in appetite/weight	Yes	11 (61%)	9 (75%)	0.535
	No	7 (39%)	3 (25%)	
insomnia/hypersomnia	Yes	14 (74%)	7 (70%)	>0.99
	No	5 (26%)	3 (30%)	
psychomotor agitation/retardation	Yes	9 (53%)	7 (70%)	0.447
	No	8 (47%)	3 (30%)	
low self-esteem	Yes	11 (61%)	6 (67%)	0.793
	No	7 (39%)	3 (33%)	
difficulty in concentration/indecision	Yes	8 (47%)	7 (64%)	0.52
	No	9 (53%)	4 (36%)	

Accordingly, we can rule out the possibility that the sex-specific associations we observed at the symptom level are due to unbalance distribution of symptomatic features in males and females from our cohort. We believe this analysis further supports our interpretation suggesting that the expression of MDD and its clinical manifestations results from transcriptional and molecular changes affecting brain regions differently in males and females.

6. Moreover, I would suggest to study symptoms profile using more “naturalistic” approaches. Some symptoms are very correlated and it has been described networks of symptoms. These networks could differ across gender too. For sample size issue, it is probably not possible to study network of symptoms here, but I would suggest to study cluster of symptoms rather than isolated symptoms.

We thank the reviewer for this suggestion. As the reviewer highlighted, it is impossible to construct networks of symptom in our cohort due to its sample size. We don't have enough case samples to cover sufficiently each symptom category and ultimately for any type of network nor even cluster analyses. The post-mortem interview process through which the clinical information was gathered also limits the assessment of the clinical manifestations and their intensity. Thus, although we appreciate the relevance of this comment, we simply cannot perform such analyses. However, as it was suggested by this reviewer, we think the data presented in our study argue in favor of such analysis. We included a couple of sentences in the discussion on this limitation (lines 536-544).

7. Another limitation deserves also more comments, if specific analysis could not be done: symptoms analysis includes symptoms like “sleep disorder” (ie hypersomnia/insomnia) that describe very different symptoms that could be associated with different biological processes. There is no specific reason that insomnia and hypersomnia are supported by the same biological abnormalities. As a consequence, results described in the manuscript should be interpreted with caution.

This is right. As mentioned in the comment above, the post-mortem nature of our study limits the assessment of the clinical manifestations and their intensity. Given we cannot perform more analysis to clarify this point, we further discussed these limitations in the discussion section as recommended (lines 542-549).

8. Please revise the gene name overall the manuscript, in human it is use to be in capital letter.

The gene names were modified as recommended.

Reviewer #3 (Remarks to the Author):

In this elegant study by Mansouri and colleagues, the authors use gene expression data from six cortical and subcortical regions of the human brain to identify associations between transcriptional profiles and MDD symptoms in males and females. Using Weighted Gene Co-expression Network Analysis (WGCNA) on RNA sequencing data from human postmortem brain tissue, the authors identify brain region specific gene networks that associate with MDD in males and females. In a follow-up analysis, they also identify gene modules that associate with specific MDD symptoms based on biserial correlations between the module eigengene and symptom expression. Using this sophisticated analysis approach, the study identifies common and sex-specific gene expression patterns and networks across six brain regions that associate with the expression of specific MDD symptoms. The data presented in this paper are highly relevant for our understanding of molecular mechanisms that could potentially be underlying neuropathology in MDD patients and provide new leads for future genetic studies in humans or functional follow-up studies in rodent models. I only have a few comments/requests for clarification.

1. More than half of women with MDD included in this study had a report of childhood abuse (13/25 MDD women vs 3/22 CTRL women). It would be interesting to know how much of the gene expression patterns are driven by a history of childhood abuse, and how the higher levels of childhood abuse in the MDD women may be related to some of the reported sex differences (also taking into account that reports of childhood abuse were largely the same between CTRL males and MDD males).

This is a very good point. Our main objective in this study was to determine the transcriptional signatures underlying the expression of specific symptoms across brain regions in males and females with MDD. Accordingly, we controlled statistically for every potential source of unwanted variance including the one from a past history of child abuse. The transcriptional impact of child abuse across brain regions in males and females with MDD is a project that other members in my lab are currently investigating.

2. This reviewer could not find the cause of death in the supplementary tables. Did MDD subjects included in this study die of suicide or natural causes or accidents?

This is an important point that was also raised by Reviewer 2. The information on cause of death was not included in the original submission. As recommended, we modified Supplementary Table 1 to include this information in the revised version of our manuscript.

3. Supplementary Table 1 shows that, as expected, a majority of MDD females was medicated. However, it is not clear which medications were taken by the subjects, if subjects were medicated at the time of death, and how medication could have influenced the gene expression findings of this study.

This is an important point. The information on medication, especially its type, duration and its compliance at the time of death was sparse for most of the sample included in this study. Nevertheless, we still examined the contribution of medication on gene expression through our principal component analysis. Since medication was not significantly associated with changes in gene expression, its effect was not adjusted for in our statistical models. This being said, as explained above, we also used RUVseq, to quantify and control for any remaining sources of variation, including medication, that was not appropriately accounted for in our statistical model. We believe this approach allowed us to account for the effect of medication that was not originally detected through our PC analysis due to a lack of clinical information but that may still have imposed an important role on overall gene expression.

4. The final sentence of the Discussion suggests that “Dissecting the transcriptional signatures underlying the expression of clinical manifestations [...] will further improve our capacity to diagnose each condition with greater precision [...]”. This statement should be further elaborated on, because based on just this sentence it is difficult to imagine how gene expression data from post-mortem brain studies can inform diagnosis or treatment of live patients in the future. Could these transcriptional data e.g. in conjunction with large-scale GWAS studies help us better understand pathology or patient stratification in the future?

We agree with the reviewer that this sentence needs further elaboration. We think that the findings from our study confirm that distinct transcriptional signatures do underly the expression of specific symptoms in males and females. We believe applying the approaches presented in our study to peripheral tissue could lead to the identification of similar signatures in easily accessible tissue. We also believe, as

suggested by the reviewer, that including information from GWAS studies, such as polygenic risks scores to this approach could allow refining the analysis to groups of patients. Together, this would provide additional tools leading toward personalized ways to diagnose and treat the disease and its clinical manifestations in males and females. We changed these sentences to reflect these thoughts.

REVIEWERS' COMMENTS

Reviewer #1 (Remarks to the Author):

All of my concerns have been addressed. Well done.

Reviewer #3 (Remarks to the Author):

The authors have addressed all my comments, and I congratulate them on a fantastic paper. CA